# Applying univariate vs. multivariate statistics to investigate therapeutic efficacy in (pre)clinical trials: A Monte Carlo simulation study on the example of a controlled preclinical neurotrauma trial

**Hristo Todorov**[1,2], **Emily Searle-White**[3], **Susanne Gerber**[1] *

**1** Faculty of Biology, Institute for Developmental Biology and Neurobiology, Johannes Gutenberg-University Mainz, Mainz, Germany, **2** Fresenius Kabi Deutschland GmbH, Oberursel, Germany, **3** Institute of Mathematics, Johannes Gutenberg-University Mainz, Mainz, Germany

* sugerber@uni-mainz.de

**Data Availability Statement:** All relevant data are within the manuscript and its Supporting Information files.

## Abstract

### Background

Small sample sizes combined with multiple correlated endpoints pose a major challenge in the statistical analysis of preclinical neurotrauma studies. The standard approach of applying univariate tests on individual response variables has the advantage of simplicity of interpretation, but it fails to account for the covariance/correlation in the data. In contrast, multivariate statistical techniques might more adequately capture the multi-dimensional pathophysiological pattern of neurotrauma and therefore provide increased sensitivity to detect treatment effects.

### Results

We systematically evaluated the performance of univariate ANOVA, Welch's ANOVA and linear mixed effects models versus the multivariate techniques, ANOVA on principal component scores and MANOVA tests by manipulating factors such as sample and effect size, normality and homogeneity of variance in computer simulations. Linear mixed effects models demonstrated the highest power when variance between groups was equal or variance ratio was 1:2. In contrast, Welch's ANOVA outperformed the remaining methods with extreme variance heterogeneity. However, power only reached acceptable levels of 80% in the case of large simulated effect sizes and at least 20 measurements per group or moderate effects with at least 40 replicates per group. In addition, we evaluated the capacity of the ordination techniques, principal component analysis (PCA), redundancy analysis (RDA), linear discriminant analysis (LDA), and partial least squares discriminant analysis (PLS-DA) to capture patterns of treatment effects without formal hypothesis testing. While LDA suffered from a high false positive rate due to multicollinearity, PCA, RDA, and PLS-DA were robust and PLS-DA outperformed PCA and RDA in capturing a true treatment effect pattern.

**Funding:** The work of HT was funded by Fresenius Kabi Deutschland GmbH. This does not alter our adherence to PLOS ONE policies on sharing data and materials. The work of SG was partly supported by the CRC 1193. ESW was supported by Center for Computational Sciences in Mainz (CSM). The funder provided support in the form of salaries for author HT, but did not have any additional role in the study design, data collection and analysis, decision to publish, or preparation of the manuscript. This does not alter our adherence to PLOS ONE policies on sharing data and materials. The specific roles of all authors are articulated in the 'author contributions' section.

**Competing interests:** The work of HT was funded by Fresenius Kabi Deutschland GmbH. This does not alter our adherence to PLOS ONE policies on sharing data and materials. The work of SG was partly supported by the CRC 1193.

## Conclusions

Multivariate tests do not provide an appreciable increase in power compared to univariate techniques to detect group differences in preclinical studies. However, PLS-DA seems to be a useful ordination technique to explore treatment effect patterns without formal hypothesis testing.

## Introduction

The aim of controlled preclinical studies is usually to investigate the therapeutic potential of a chemical or biological agent, or a certain type of intervention. For this purpose, animals are randomized to a control group and a number of treatment groups in a manner similar to clinical trials. For quantitative endpoints, treatment effects are evaluated by assessing mean differences between control and intervention groups. In an effort to obtain as much information as possible with minimal cost of life, usually multiple endpoints are included in the trial [1], which is further motivated by the fact that the optimal efficacy endpoint for a specific disease might not be known. In this context, the null hypothesis of no treatment effect ($H_0$) can be rejected in two ways. The standard approach consists of performing independent univariate tests on each variable separately. However, this strategy might lead to an inflated family-wise error rate. In addition, different endpoints are usually correlated, implying that preclinical trials are multi-dimensional in nature. Consequently, the second approach is to use a multivariate technique, which accounts for the covariance/correlation structure of the data. $H_0$ is usually tested on some kind of linear combination of the original variables. Due to the increased complexity of analysis and interpretation of results in this case, such an approach has found limited use in preclinical research so far.

A number of studies have highlighted the potential benefits of multivariate techniques in the context of preclinical trials [2] and more specifically animal neurotrauma models [3–7]. Traumatic or ischemic events to the central nervous system such as stroke, spinal cord or traumatic brain injury are followed by a multi-faceted pathophysiology which manifests on molecular, histological and functional levels [8–11]. Individual biological mechanisms that are disrupted by or result from the neurotrauma such as apoptosis [12, 13], neuroinflammation [14–18], oxidative stress [18–20] and plasticity alterations [21, 22] have provided therapeutic targets in animal models. However, translation of candidate therapies to humans continues to be mostly unsuccessful [23–26]. Many studies indicate that individual biological processes interact together in determining functional outcome, which is why multivariate measures might capture the complex disease pattern more successfully and therefore detect therapeutic intervention efficacy with increased sensitivity [3, 4]. However, no solid proof of the superiority of multivariate methods beyond these theoretical considerations has been ascertained so far.

The aim of our current study was to obtain empirical evidence as to whether univariate or multivariate statistical techniques are better suited for detecting treatment effects in preclinical neurotrauma studies. For this purpose, we performed simulations under a broad range of conditions while simultaneously trying to mimic realistic experimental conditions as closely as possible. We investigated the empirical type I error rate as well as empirical power of several competing techniques and evaluated factors which impact their performance.

## Methods

### Simulation procedure

We performed a Monte Carlo study using the statistical software R [27] and following recommendations of Burton et al. for the design of simulation studies [28]. Artificial data were based

on a real study in a rat model of traumatic brain injury. In the preclinical trial, twenty animals per group received either vehicle control or a therapeutic agent. Functional outcome was evaluated based on 6 different endpoints including 20-point neuro-score, limb placing score, lesion and edema volume, and T2 lesion in the ipsilateral and contralateral hemisphere. All variables were measured repeatedly on three time points, therefore resulting in a data matrix with 18 columns. In order to obtain more general estimates of the mean vector and covariance matrix for subsequent simulations, a non-parametric bootstrap procedure was applied using the data from the saline control group from the *in vivo* study. Since two animals from this group were excluded from the study, the resampling procedure was conducted with the available 18 animals. 10,000 samples were drawn from the original data with replacement and the average mean vector and covariance matrix were then calculated. In order to retain the covariance structure of the data, complete rows of the data matrix (corresponding to all measurements from a single animal) were always sampled as a 18x1-dimensional vector. The *nearPD* R function was then employed to force the calculated dispersion matrix to be positive definite. The resulting mean vector and covariance matrix were used as parameters for multivariate distributions, from which data for subsequent simulations were sampled (see S1 Appendix of Tables 1 and 2). We generated one control group and three treatment groups under each scenario, which corresponds to a typical preclinical trial design where increasing doses of a therapeutic agent are tested against a control treatment.

## Simulation factors

**Sample size.** We performed simulations with 5, 10, 15 and 20 measurements per treatment group to investigate the impact of sample size. These values were selected to represent realistic group sizes commonly encountered in preclinical trials. Additionally, we performed simulations with 30, 40 and 50 replicates per group to investigate the effect of a larger sample size beyond those typical for animal studies. In the course of this study we use the terms measurements, subjects and replicates per group interchangeably.

**Effect size.** Treatment effects were based on Cohen's d with values set to 0, 0.2, 0.5 and 0.8 corresponding to no effects, small, moderate and large statistical effect sizes relative to the control group, respectively [29]. We chose Cohen's d because this standardized statistical measure of effect size is independent of the scale of the original variables. The population mean values for the treatment groups were then calculated using the formula $\mu_1 = \mu_0 \pm s^* d$, where $\mu_0$ corresponds to the population mean of the respective variable in the control group and $s$ signifies the standard deviation of both groups in case of equal variance or the average standard deviation in case of unequal variance. We performed simulations with no treatment effects in all groups to investigate empirical type I error rate. Additionally, we investigated empirical power by simulating either large, moderate or small effects in the treatment groups relative to the control group.

**Distribution of dependent variables.** The dependent variables were simulated to follow a multivariate normal distribution to comply with the assumptions of the investigated methods. Additionally, we employed the multivariate lognormal distribution and the multivariate gamma distribution in order to investigate the impact of departures from normality. The multivariate gamma distribution was modelled using its shape parameter α and its rate parameter β. These parameters were derived from the target mean and variance values using the following relationships: $\mu = \alpha/\beta$ and $\sigma^2 = \alpha/\beta^2$, where μ and $\sigma^2$ correspond to the mean and variance of the gamma distribution, respectively. Since we wanted to simulate specific values for the mean and variance, we used the following equations to obtain the shape and rate parameter of

the gamma distribution: $\alpha = \mu^2 / \sigma^2$ and $\beta = \mu / \sigma^2$. The correlation matrix used for the simulation of multivariate data sets is shown in S1 Appendix of Table 2.

**Variance.**   Parametric univariate methods to detect mean differences assume that variance in all groups is equal, which in the multivariate case extends to the assumption of homogeneity of covariance matrices [30]. Therefore we first performed simulations with all groups having equal variance. Then we simulated treatment groups having variance twice or 5 times higher than the variance in the control group. This allowed us to investigate the impact of increasing variance heterogeneity.

Factors were crossed to produce 252 different simulation scenarios with 1000 replicate data sets generated under each combination of simulation conditions.

## Methods to detect treatment effects

**Univariate statistics.**   The univariate approach of investigating treatment differences between groups consisted of a series of independent analysis of variance (ANOVA) tests on each outcome variable separately. Furthermore, we applied Welch's ANOVA as implemented in the *oneway.test* R function, which does not assume equal variance between groups [31]. In order to take the repeated measures nature of the input data into account, we also performed linear mixed effects tests for each endpoint. Since we did not simulate an interaction between treatment effect and time, we only included the main effects in the mixed effects model without an interaction term. We rejected $H_0$ of no treatment effect if the main effect for the treatment factor was significant.

**Multivariate statistics.**   The first multivariate strategy we investigated was performing ANOVA tests on principal component (PC) scores obtained from the original variables. We used eigen decomposition of the population correlation matrix in order to calculate the PCs, which is the preferred approach when variables are measured on different scales [30, 32]. Based on the Kaiser criterion, we only retained components whose corresponding eigenvalue was greater than one [33]. Component scores were obtained by multiplying the standardized data matrix of original variables with the eigenvectors of the population correlation matrix [32].

The second multivariate technique consisted of a series of multivariate analysis of variance (MANOVA) tests on each study variable with repeated measures. Each repeated measure was considered a separate dependent variable for the respective MANOVA. Thus, we performed 6 MANOVA tests, each of which included the three repeated measures of one endpoint as the dependent variables. The significance of the MANOVA tests was evaluated using four different statistics which are commonly provided by statistical software such as R, SAS or SPSS: Wilks' lambda [34], Lawley-Hotelling trace [35], Pillai's trace [36] and Roy's largest root [37].

In all cases, $H_0$ was rejected when the p-value from the omnibus test was less than 0.05; no specific contrasts or post hoc analyses were considered. Different techniques were evaluated based on the empirical type I error rate or on empirical power. Empirical type I error rate was defined as the number of significant statistical tests divided by the total number of tests when no treatment effects were simulated. Empirical power was defined as the number of significant tests divided by the total number of tests in the cases when treatment effects were simulated.

## Multivariate dimensionality reduction techniques for pattern analysis

In addition to formally comparing the type I error rate and power of univariate and multivariate statistics, we also investigated if ordination techniques might be useful to detect patterns of treatment effects in multi-dimensional preclinical data sets. We focused on methods that perform ordination and dimensionality reduction based on Euclidean distances and are therefore suitable for quantitative and semi-quantitative data. First, we applied PCA, linear discriminant

analysis (LDA), redundancy analysis (RDA), and partial least squares discriminant analysis (PLS-DA) on 1000 simulated data sets with 5 measurements per group and no treatment effects. We plotted the first versus the second multivariate dimension and visually inspected the plots. If the 95% confidence ellipse around the control group did not overlap with the confidence ellipses around the data points for the treatment groups, we considered that the ordination method falsely captured a treatment effect pattern in the data. Next, we examined the sensitivity of the ordination methods to detect true treatment effect patterns by simulating 1000 data sets with 5 measurements per group and huge treatment effects (Cohen's d = 2.0). We used this effect size as we did not observe a difference between groups when smaller effect sizes were simulated. We considered that the respective method correctly accounted for a treatment effect pattern in the data if the 95% confidence ellipse around the control group did not overlap with the confidence ellipses around the simulated treatment groups.

Finally, we provide an applied example of combining dimensionality reduction techniques with formal hypothesis testing using one simulated data set with 5 measurements per group and treatment effects on only half of all the variables.

## Results

### Competing multivariate statistics

Prior to investigating the performance of univariate and multivariate techniques, we examined the four MANOVA test statistics in order to identify the most appropriate for subsequent comparisons. Fig 1 shows representative results for the type I error and power of the

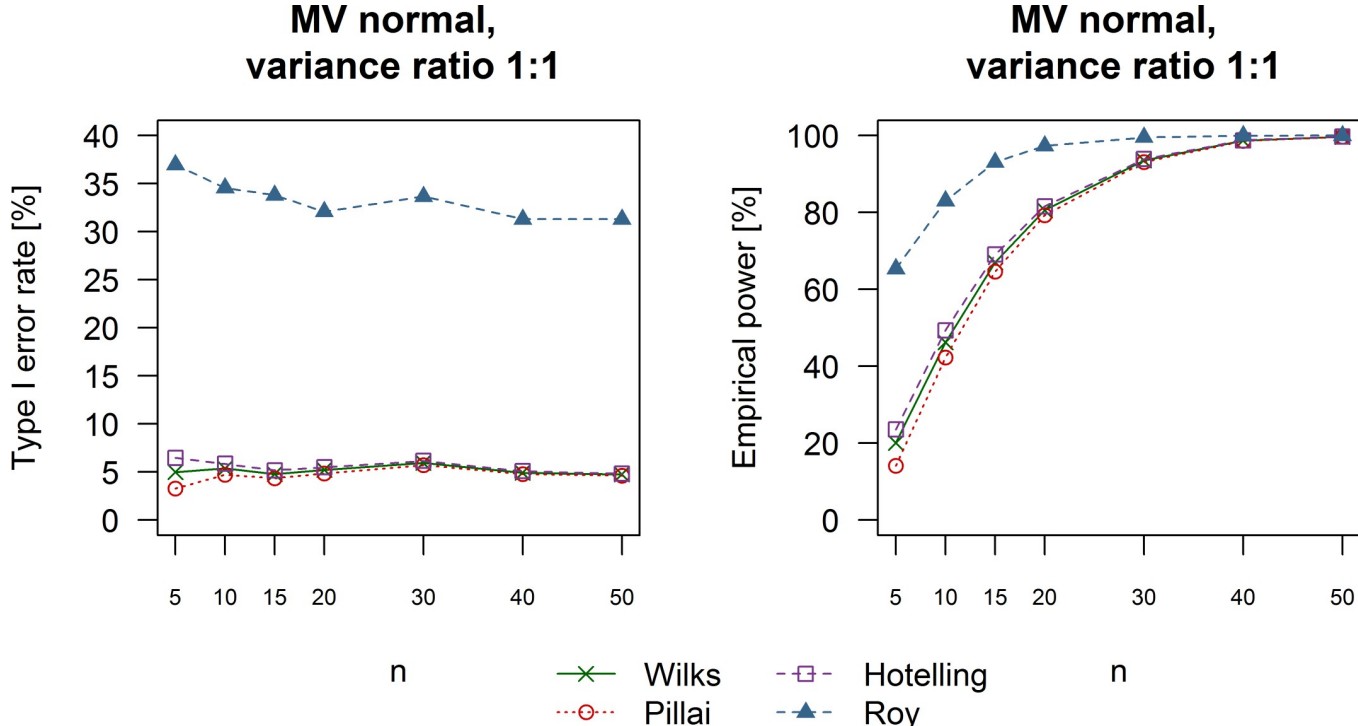

**Fig 1. Performance of different multivariate statistics.** Example plots show empirical type I error and power of the MANOVA test using four common multivariate test statistics. Type I error rate is shown for the simulation scenario with no treatment effects, equal variance in all groups and data drawn from a multivariate normal distribution. An example of power analysis is shown for a simulation with large treatment effects (Cohen's d equal to 0.8), equal variance in all groups and data sampled from a multivariate normal distribution. Hotelling: Lawley-Hotelling trace; Pillai: Pillai's trace; Roy: Roy's largest characteristic root; Wilks: Wilks' lambda.

MANOVA test (see S1 Appendix of Figs 1–4 for complete results) using the four different statistical criteria. We observed the same trend under all simulation scenarios with Roy's largest root having a considerably high false positives rate over 30%. In contrast, the remaining statistics exhibited very similar type I error rates. Pillai's trace was the most robust measure followed by Wilks' lambda and Lawley-Hotelling trace. Roy's largest root was not considered with regards to power analysis due to the unacceptably high type I error rate. Pillai's trace consistently demonstrated the lowest power. In contrast, Wilks' lambda was associated with a slightly higher probability of correctly rejecting the null hypothesis in the presence of treatment effects than Pillai's trace but it was outperformed by Lawley-Hotelling trace. However, we chose Wilks' lambda for further analysis because it provided a good compromise between type I error rate and power in comparison to the other multivariate test statistics.

### False positive rate

Empirical type I error rates of the methods we evaluated under different simulation scenarios are summarized in Fig 2. Differences between univariate and multivariate methods were negligible under all simulation conditions. Furthermore, all methods managed to remain close to the nominal level of type I error rate around 5% even in the case of extreme variance heterogeneity (variance ratio between control and treatment group equal to 1:5). Interestingly, Welch's ANOVA was associated with a slightly higher false positive rate compared to other methods when data were sampled from a multivariate lognormal distribution combined with extreme variance heterogeneity. Furthermore, linear mixed effects models had a slightly higher type I error rate in the case of 5 subjects per group.

### Empirical power

The results we obtained for empirical power under different simulation conditions are depicted in Figs 3–5. Linear mixed effects models outperformed the remaining methods in the case of variance equality or moderate variance heterogeneity (variance ratio 1:2) with smaller sample sizes of 5 to 20 subjects per group regardless of the effect size we simulated. Welch's ANOVA was as powerful as regular ANOVA when the variance between the control and treatment groups was equal. Furthermore, Welch's ANOVA outperformed all other methods when we simulated moderate or small effect sizes combined with extreme variance heterogeneity (ratio of 1:5 between the control and treatment groups) and data coming from multivariate lognormal or gamma distributions. MANOVA tests were slightly more powerful than the two types of ANOVA in the cases of equal variance but still failed to outperform linear mixed effects models under these simulation scenarios. The multivariate strategy of ANOVA tests on PCA scores was universally associated with the lowest rate of rejecting $H_0$. It is also worth mentioning that adequate levels of power of around 80% were achieved in the case of at least 20 measurements per group and large treatment effects (Cohen's d equal to 0.8, Fig 3). Simulating moderate treatment effects (Cohen's d equal to 0.5, Fig 4) required a sample size of at least 40 replicates per group in order to achieve levels of power of around 80% Finally, the rate of rejecting $H_0$ varied between 5% and 25% when we simulated small treatment effects (Cohen's d equal to 0.2, Fig 5).

### Comparison of ordination techniques for pattern analysis of treatment effects

We investigated if the dimensionality reduction techniques LDA, PCA, RDA, and PLS-DA could be useful for investigating patterns of treatment effects without formal hypothesis testing. In 1000 simulated data sets without treatment effects and 5 measurements per group, we counted how often the control group was separated from treatment groups along the first and

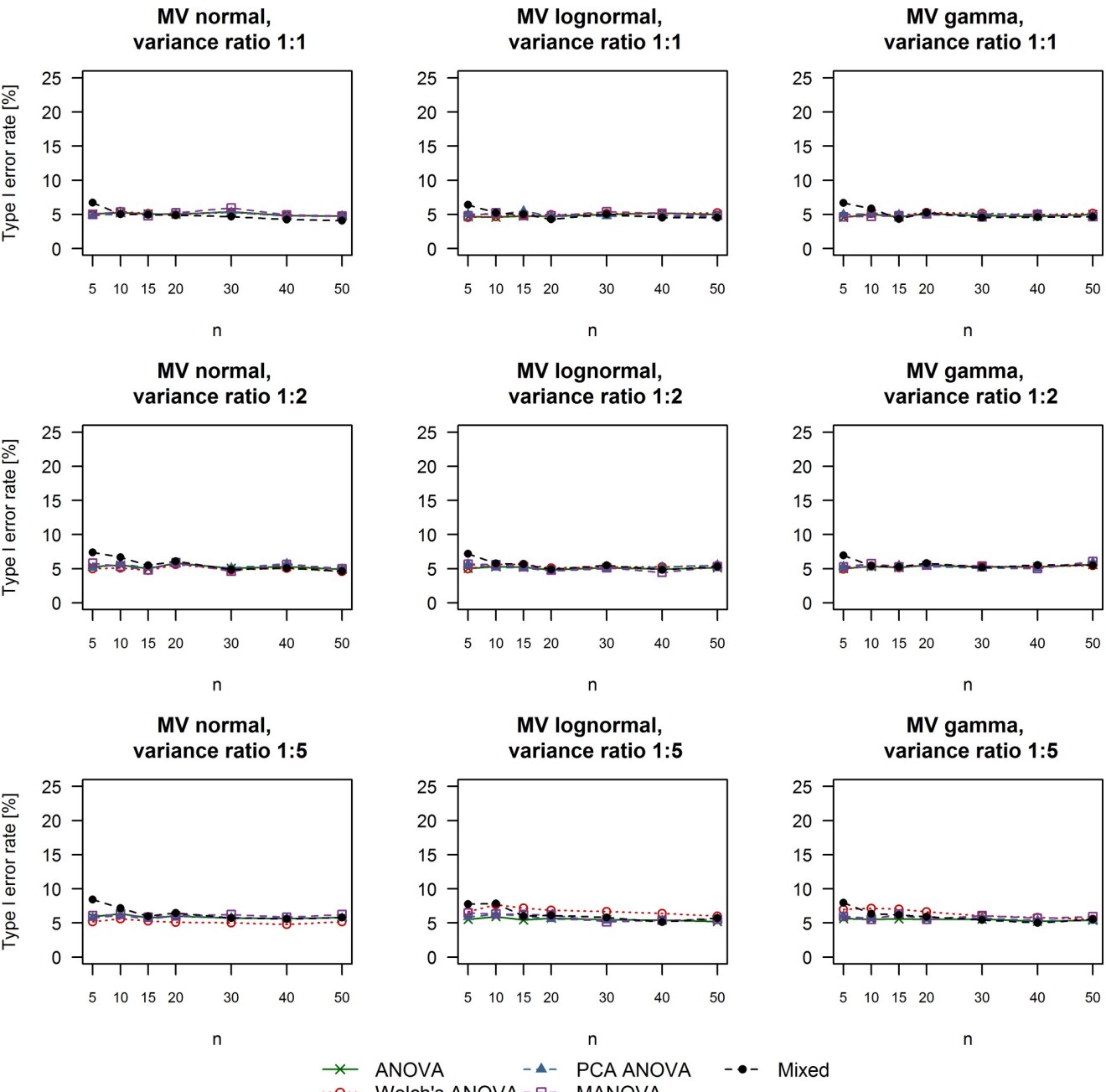

**Fig 2. Type I error rate of univariate and multivariate techniques under different simulation conditions.** The title of each plot reports the multivariate distribution from which the data were sampled as well as the variance ratio between the simulated control and treatment groups. ANOVA: Analysis of variance; MANOVA: Multivariate analysis of variance; Mixed: Linear mixed effects model; MV: Multivariate; PCA: Principal component analysis.

second multivariate dimensions (indicated by non-overlapping 95% confidence ellipses). LDA captured a false treatment effect pattern in 387 cases corresponding to a false positive rate of 38.7%. In contrast, the control group was not separated from treatment groups in any of the simulated sets when using PCA, PLS-DA, or RDA for dimensionality reduction. Example plots are shown in Fig 6 (the whole set of plots is available in S2 Appendix). Due to the unacceptably high false positive rate, we did not further consider LDA. Next, we simulated 1000

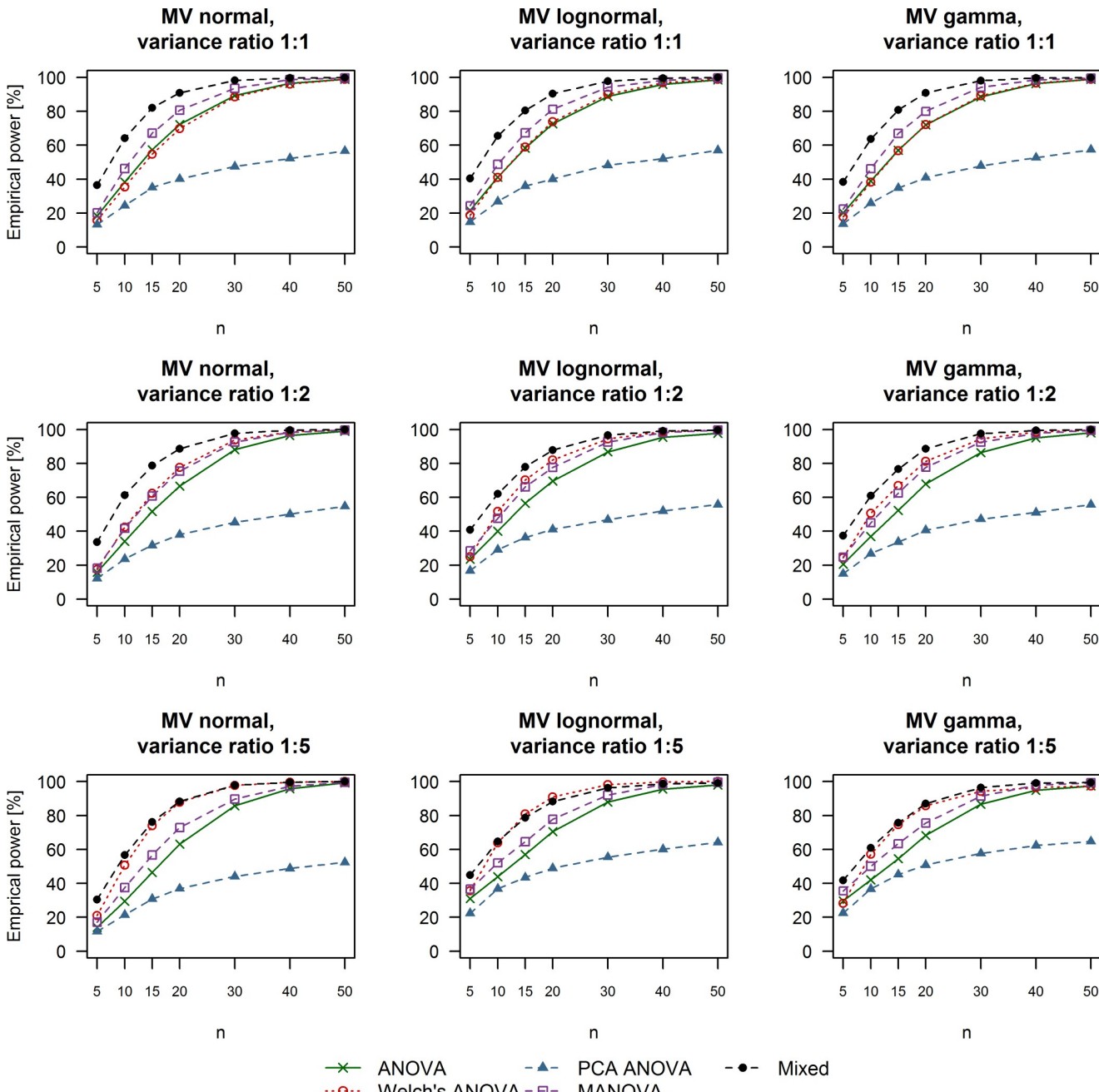

**Fig 3. Empirical power of univariate and multivariate techniques in case of large treatment effects (Cohen's d equal to 0.8).** The multivariate distribution from which the data were drawn as well as the variance ratio between simulated control and treatment groups are summarized in the title of each respective plot. ANOVA: Analysis of variance; MANOVA: Multivariate analysis of variance; Mixed: Linear mixed effects model; MV: Multivariate; PCA: Principal component analysis.

data sets with huge treatment effects (Cohen's d equal to 2.0) with 5 measurements per group and investigated how often the control group was separated from treatment groups in reduced multivariate space. PLS-DA managed to capture the true treatment pattern in 13.8% of the cases whereas PCA only separated the control from treatment groups in 7.7% of the simulations. RDA only marginally outperformed PCA and reported a true treatment effect pattern in 9.6% of the cases (the complete simulated set of plots is available in S3 Appendix).

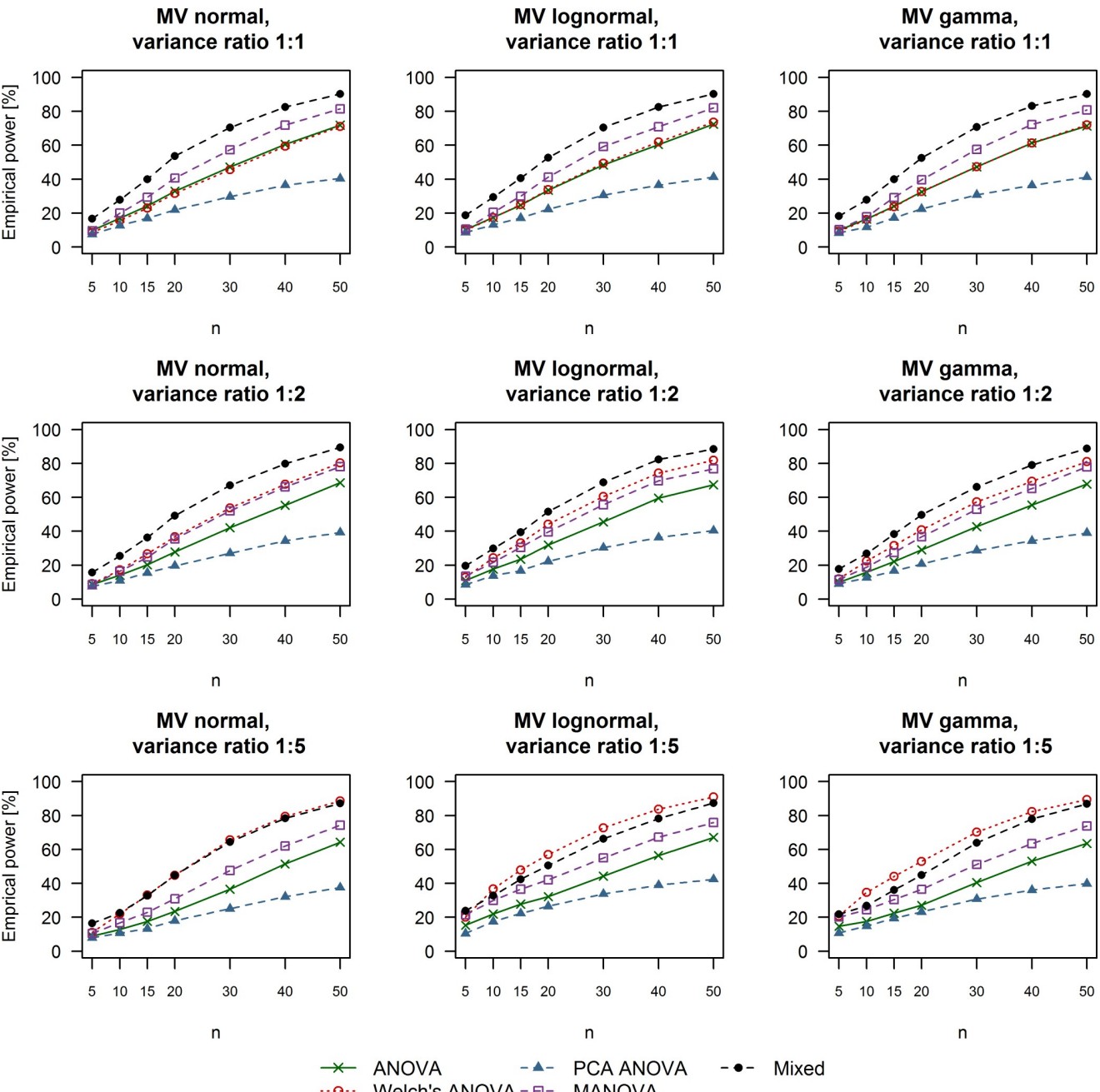

**Fig 4. Empirical power of univariate and multivariate techniques in case of moderate treatment effects (Cohen's d equal to 0.5).** The multivariate distribution from which the data were drawn as well as the variance ratio between simulated control and treatment groups are summarized in the title of each respective plot. ANOVA: Analysis of variance; MANOVA: Multivariate analysis of variance; Mixed: Linear mixed effects model; MV: Multivariate; PCA: Principal component analysis.

## A practical example of applying ordination techniques and statistical testing methods

In order to give an example of how ordination techniques can be combined with statistical testing methods in practice, we simulated a data set with 5 variables per group and huge treatment effects for 9 out of the total 18 variables which we randomly selected. The endpoints with

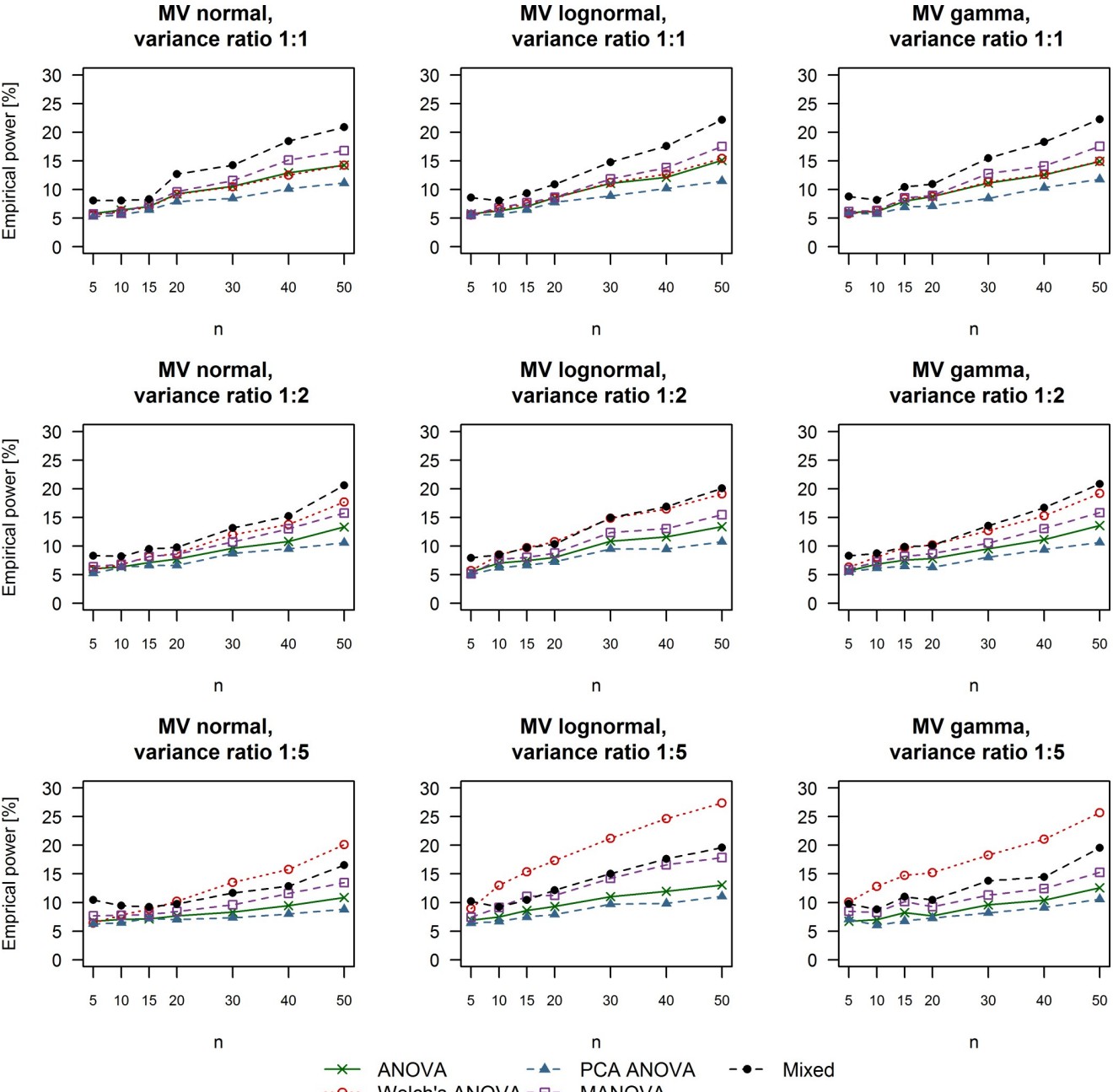

**Fig 5. Empirical power of univariate and multivariate techniques in case of small treatment effects (Cohen's d equal to 0.2).** The multivariate distribution from which the data were drawn as well as the variance ratio between simulated control and treatment groups are summarized in the title of each respective plot. ANOVA: Analysis of variance; MANOVA: Multivariate analysis of variance; Mixed: Linear mixed effects model; MV: Multivariate; PCA: Principal component analysis.

simulated treatment effects were 20-point neuroscore on day 1 and day 7, limb placing score on day 1 and day 7, lesion volume on day 1 and day 7, edema volume on day 1 and day 14 and T2 lesion in the contralateral cortex on day 1. The remaining 9 variables were drawn from the same distributions in the control and the 3 treatment groups without simulated treatment effects.

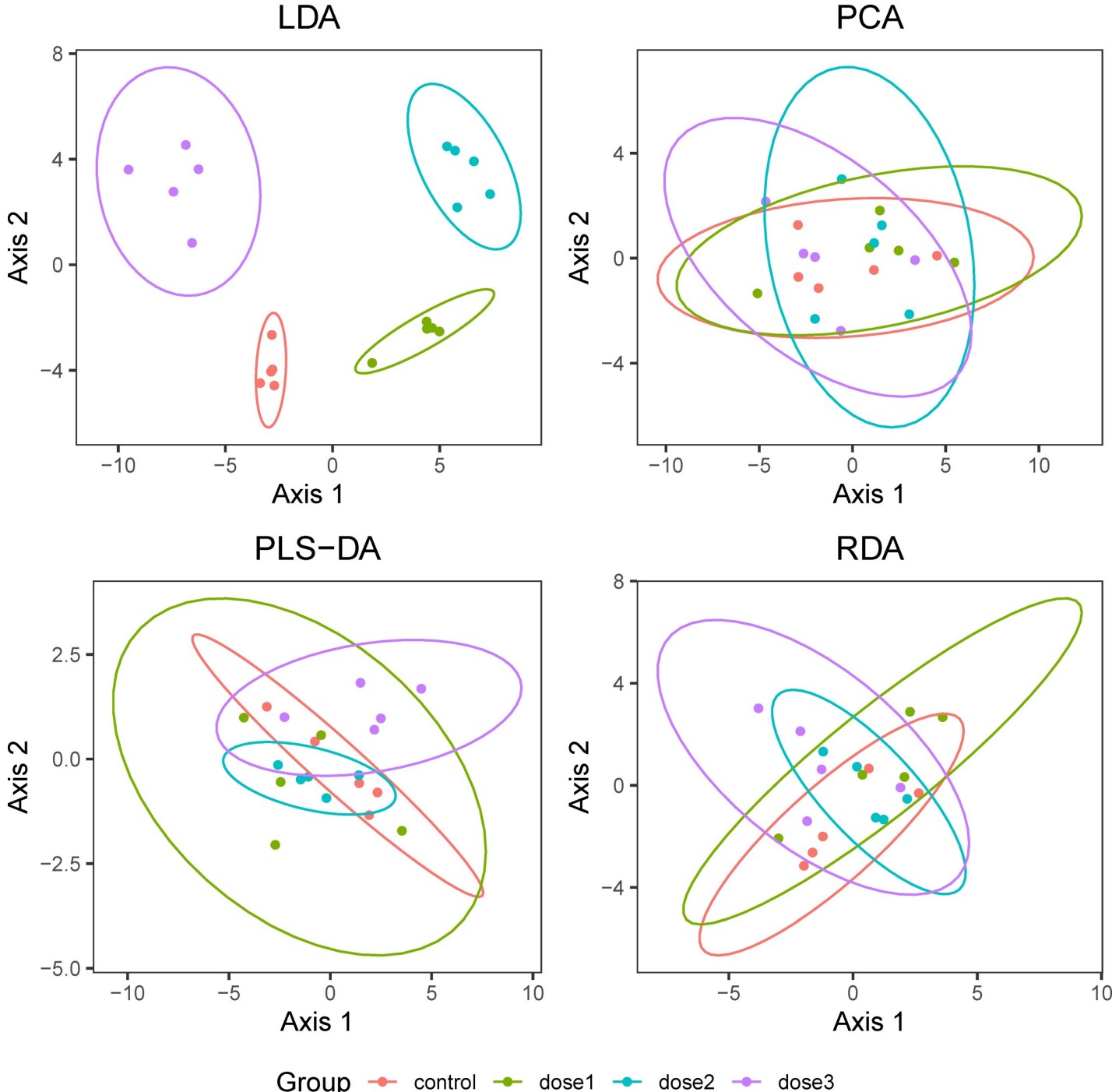

**Fig 6. Comparison of ordination techniques for pattern analysis in the case when no treatment effects were simulated.** Plots show results for one out of 1000 simulations with 5 measurements per group drawn from a multivariate normal distribution with equal variance between control and treatment groups. The ordination technique was considered to falsely capture a treatment effect pattern in the data in case of non-overlapping 95% confidence ellipse of the control group with the confidence ellipses for the treatment groups (dose1 to dose3). LDA: Linear discriminant analysis; PCA: Principal component analysis; PLS-DA: Partial least squares discriminant analysis; RDA; Redundancy analysis.

In the first step of the analysis, we applied PLS-DA which was the most sensitive technique in our simulations to investigate if the control group differed from the treatment groups in

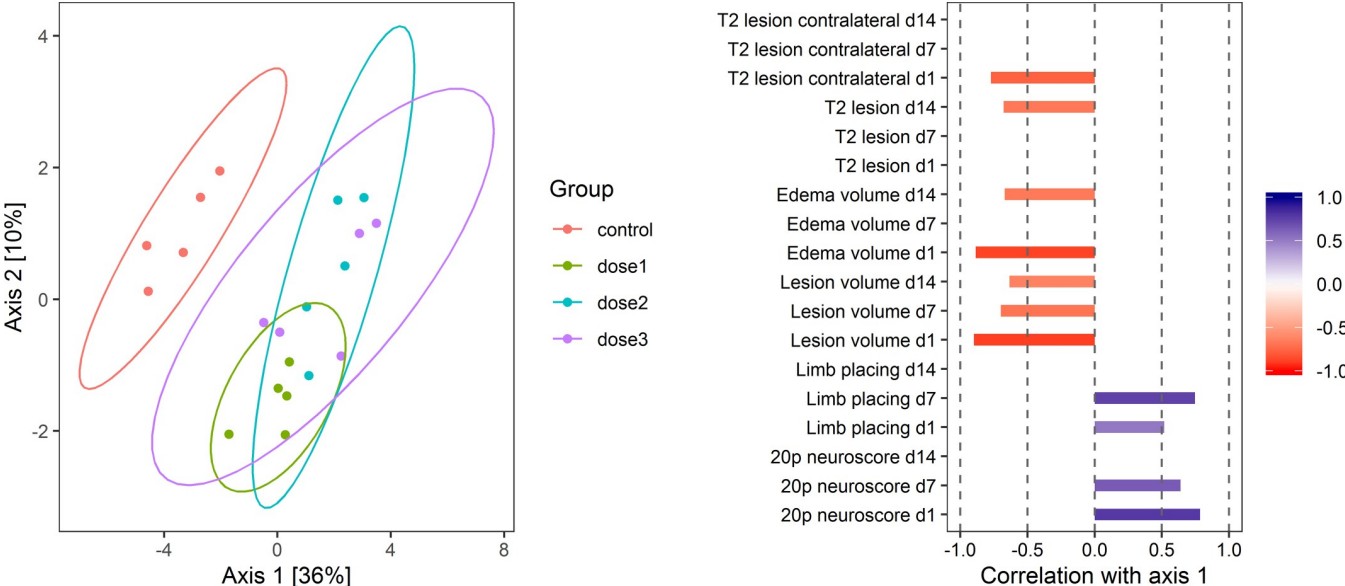

**Fig 7. Partial least squares discriminant analysis (PLS-DA) to identify treatment effect patterns.** We simulated a data set with 5 measurements per group and huge treatment effects for 9 randomly selected endpoints out of the 18 variables in the data set. The control group was separated from the treatment groups along the first multivariate dimension in the PLS-DA analysis We calculated the correlation of the original variables with this dimension to identify which original endpoints explained the multivariate pattern. Correlations with an absolute value below 0.5 were set to 0 in order to filter out unimportant variables. All 9 variables with simulated treatment effects were significantly correlated with the first multivariate axis. Two additional variables without simulated treatment effects (lesion volume at day 14 and T2 lesion at day 14) were also significantly correlated with the first multivariate axis.

reduced multivariate space. We observed that the control group was separated from the treatment groups along the first multivariate axis which accounted for 36% of the variance (Fig 7). In order to investigate which of the original variables are responsible for group separation, we calculated the correlations of the original variables with the first PLS-DA multivariate dimension (axis 1) along which the control and treatment groups were separated. Correlations with an absolute value below 0.5 were set to 0 in order to filter out unimportant variables. The correlation pattern indicated that all variables with simulated treatment effects along with two additional variables (lesion volume at day 14 and T2 lesion at day 14) contributed to the separation of the control from the treatment groups. Therefore, PLS-DA managed to capture the treatment effect pattern by identifying all original variables with simulated treatment effects as important for group separation in reduced space.

Next, we followed up on the multivariate pattern analysis by performing statistical testing with linear mixed effects models for each variable with repeated measures. The interaction term between treatment and time was highly significant for all six endpoints thereby rejecting $H_0$ of no treatment effects even for T2 lesion, which was the only variable without any simulated treatment effects at any time point. Next, we performed post-hoc analysis comparing the treatment groups against the control group for each time point separately. Results are shown in Table 1. The difference for the 20-point neuroscore was significant only between treatment groups 2 and 3 compared to the control group and no statistically significant difference was detected for 20-point neuroscore at day 7. Similarly, post-hoc analysis did not detect a treatment effect for any of the groups for lesion volume at day 7 and edema volume at day 14. In contrast, all treatment effects were identified for lesion volume at day 1, edema volume at day 1 and T2 lesion in the contralateral cortex at day 1.

The difference between the control and treatment groups 2 and 3 for T2 lesion at day 14 was reported as significant even though we did not simulate treatment effects for this variable.

**Table 1. Post-hoc analysis following linear mixed effects models for variables with repeated measures.**

| Variable | Control vs. dose 1 | Control vs. dose 2 | Control vs. dose 3 |
|---|---|---|---|
| **20 point neuroscore day 1** | 0.51 | **0.0001** | **0.0027** |
| **20 point neuroscore day 7** | 0.293 | 0.095 | 0.074 |
| 20 point neuroscore day 14 | 0.688 | 0.593 | 0.354 |
| **Limb placing score day 1** | 1.000 | 0.483 | **0.047** |
| **Limb placing score day 7** | 0.298 | **0.033** | **0.047** |
| Limb placing score day 14 | 0.297 | 0.483 | 0.383 |
| **Lesion volume day 1** | **<0.0001** | **<0.0001** | **<0.0001** |
| **Lesion volume day 7** | 0.119 | 0.117 | 0.131 |
| Lesion volume day 14 | 0.778 | 0.332 | 0.487 |
| **Edema volume day 1** | **0.0002** | **<0.0001** | **<0.0001** |
| Edema volume day 7 | 0.266 | 0.494 | 0.824 |
| **Edema volume day 14** | 0.129 | 0.122 | 0.087 |
| T2 lesion day 1 | 0.309 | 0.338 | 0.453 |
| T2 lesion day 7 | 0.826 | 0.627 | 0.827 |
| T2 lesion day 14 | 0.203 | **0.001** | **0.05** |
| **T2 lesion contralateral cortex day 1** | **0.004** | **0.0005** | **<0.0001** |
| T2 lesion contralateral cortex day 7 | 0.230 | 0.286 | 0.316 |
| T2 lesion contralateral cortex day 14 | 0.828 | 0.201 | 0.529 |

We performed linear mixed effects analysis for each endpoint with repeated measures followed by post-hoc pairwise comparisons between the control and each treatment group for each time point separately. Variables with simulated treatment effects are highlighted with a bold font. The p-values from the post-hoc comparisons are reported in the table. P-values less than 0.05 are highlighted with a bold font.

Altogether, post-hoc analysis following linear mixed effects models captured most but not all individual differences between the control and treatment groups. In contrast, the multivariate pattern analysis using PLS-DA marked all variables with simulated treatment effects as important for group separation in reduced multivariate space.

## Discussion

Using Monte Carlo simulations, we evaluated the performance of a number of univariate and multivariate techniques in an effort to identify the most optimal strategy for detecting treatment effects in preclinical neurotrauma studies.

Importantly, type I error rate was not drastically inflated beyond the 5% nominal rate for all hypothesis testing methods under the simulation scenarios we investigated, even when assumptions of normality and homogeneity of variance were violated. Nevertheless, we only simulated a maximal variance inequality ratio of 1:5 between control and treatment group. Moreover, sample size was always equal. Extreme heterogeneity is more problematic in case of unequal group sizes especially when the smallest group exhibits the largest variance [38]. In such cases, a variance-stabilizing transformation such as log-transformation of the response variables is advisable. Alternatively, in the univariate case, a non-parametric technique might be used (e.g. Friedman or Kruskal-Wallis test). In case that MANOVA is performed, a more robust statistic might be chosen. Our results suggest that Pillai's trace would be the most appropriate under these conditions.

In terms of power, taking the repeated measures nature of the data into account proved to be the optimal strategy as linear mixed effects models outperformed the other methods when variance between groups was equal or when variance heterogeneity was moderate. Linear

mixed effects are a flexible class of statistical methods which allow building models of increasing complexity with different combinations of random intercepts and slopes. In practice, however, it might be challenging to assess the significance of fixed effects in the model based on F-tests as the degrees of freedom might not be correctly estimated. In our current study, we used the Kenward-Roger approximation for determining the degrees of freedom [39]. Alternatively, likelihood ratio tests might be used in order to test if including the factor of interest significantly improves the model fit compared to a model without the specific factor. Importantly, this requires refitting the linear mixed effects model using maximum likelihood to estimate parameters as usually these models are calculated using restricted maximum likelihood.

When the assumptions of normality and homogeneity of variance were violated, univariate Welch's ANOVA tests outperformed the remaining methods especially with small effect sizes. Furthermore, the rate of rejecting $H_0$ was equivalent to that of standard ANOVA when data were sampled from a multivariate normal distribution with equal variance between groups. These results suggest that Welch's ANOVA might be more appropriate for statistical testing of treatment effects than the much more popular standard ANOVA F-test. Additionally, univariate methods offer the advantage of directly investigating differences on endpoints of interest whereas multivariate tests are applied on a linear combination of the original variables. Nevertheless, ignoring the correlation structure of the response variables may result in misleading conclusions. Correlated variables reflect overlapping variance and therefore univariate tests provide little information about the unique contribution of each dependent variable [30].

The issue of correlated outcome measures is addressed by employing multivariate methods. When differences are evaluated between groups which are known a priori, MANOVA is the technique of choice. In our study, MANOVA offered a marginally higher power than univariate ANOVAs when the assumption of variance homogeneity was met. However, a practical issue of this method is that standard software reports four different statistics which do not always provide compatible results. Under all simulation conditions we investigated, Roy's largest root was associated with an unacceptably high type I error rate. This would make interpretation of results with real high-dimensional data sets with few measurements per variable very ambiguous. However, Wilks' lambda, Lawley-Hotelling trace and Pillai's trace were robust to false positives. In agreement with previous reports, Pillai's criterion was the most conservative, which would make it more appropriate when assumptions of MANOVA are violated [40, 41]. Nevertheless, we opted to use Wilks' lambda for subsequent comparisons between different techniques because it offered similar robustness but slightly increased power. Another trade-off of MANOVA and multivariate techniques in general is the complexity of interpretation. If the omnibus test is significant, a researcher will often want to more precisely identify the variables which are responsible for group separation. Ideally, follow-up tests should retain the multivariate nature of the analysis. Such strategies include descriptive discriminant analysis [30, 42] or Roy-Bargmann stepdown analysis [30, 43].

A crucial factor we did not consider in our study is missing data which cannot be handled by multivariate statistical methods. If the degree of missingness is within a reasonable range (e.g. not more than 10%) and the assumption of missing at random is satisfied, then a multiple imputation technique might be employed to estimate the missing data from the existing measurements. Otherwise, a more flexible data analysis method must be employed such as for instance linear mixed effects models, which are able to handle missing data.

Since MANOVA only very marginally outperformed univariate ANOVAs and failed to provide an increase of power compared to linear mixed effects models, we believe that this does not offset the increased complexity and inability to handle missing data. Therefore, our results would suggest that MANOVA tests are not a practical option for formal hypothesis testing in preclinical studies with small sample sizes.

It is important to note that different methods achieved acceptable levels of power of around 80% only when we simulated large treatment effects with 20 measurements per group or moderate effects with at least 40 replicates per group. This finding highlights a serious issue not only in neurotrauma models but in preclinical research altogether, namely that typical sample sizes in animal studies do not ensure adequate power unless the effect size is large. Accordingly, some authors argue that animal studies should more closely adhere to the standards for study conduct and reporting applicable to controlled clinical trials [1, 44]. In a randomized clinical study, sample size is calculated a priori based on a specific effect size, assumptions about the variance in the response variable, and the desired level of power. In theory, the ARRIVE guidelines which were developed in order to improve the quality of study conduct and reporting of animal trials [45] as well as animal welfare authorities [46] require formal justification for sample size selection. Group size should be appropriate to detect a certain effect with adequate power while simultaneously ensuring that no more animals than necessary are used [46]. In practice, power calculations for preclinical trials are challenging for a number of reasons. For instance, information about the variance in the response variable might not be available a priori, however this issue might be tackled by performing a small scale pilot study. Another problem may be that the estimated effect is small while the variance in the selected endpoint is high, which results in such large group sizes that might not be acceptable for animal welfare regulators. One possible way to address this problem is to identify methods which are associated with higher power in small samples or try to reduce the variability in the response variables by possibly including other covariates in the analysis [47]. A recent development in the effort to increase power of animal studies includes performing systematic reviews and meta-analysis of existing studies [48]. This approach is well established in clinical research and it allows scientists to appraise estimated effect sizes more systematically and put them in the context of existing reports. The majority of preclinical meta-analyses which have been performed in the field of neurotrauma so far are related to experimental stroke (e.g. [49–54]). However pre-clinical meta-analyses on e.g. spinal cord injury [55, 56] and subarachnoid hemorrhage [57] have also been published.

However, since a meta-analysis is not always practicable, especially when a novel study is conducted, we investigated if ordination techniques might be useful to detect treatment effect patterns with small sample sizes. Multivariate techniques classically rely on data sets consisting of more observations than variables, which is not always the case in animal studies especially in the omics era. Therefore, we first evaluated if LDA, PCA, PLS-DA, or RDA falsely report non-existing patterns in simulated data sets without treatment effects. With 5 measurements per group and 18 variables, LDA was associated with a false positive rate of 38.7% while PCA, PLS-DA, and RDA did not capture false patterns in the data. The extreme over-fitting we observed for LDA is due to multicollinearity in the data set (see S1 Appendix of Table 2 for the correlation matrix used for simulating multivariate data sets) combined with a small sample size [58]. While this is not necessarily a novel finding, our simulation results highlight the dangers of carelessly applying a dimensionality reducing technique to multivariate data sets with more variables than measurements, which often leads to false inferences. In contrast, PCA is capable of overcoming the "large p, small n" problem by reducing the large number of variables to a few uncorrelated components. The method only imposes the constraint that the first component captures the direction of greatest variance in the data hyper-ellipsoid [32] and does not perform regression or classification of data. Therefore multicollinearity poses no issue. However, group assignment is ignored and so differences between groups do not necessarily become apparent in reduced space. RDA is the supervised version of PCA and it imposes the constrain that the dependent variables in reduced space are linear combinations of the grouping variable. Surprisingly, RDA demonstrated only a slightly increased sensitivity to

detect true treatment effect patterns in our simulations compared to PCA. Conversely, PLS-DA clearly outperformed both PCA and RDA. Although PLS-DA uses the quantitative variables to predict group membership similarly to classical LDA, classification is performed after dimensionality reduction [59]. PLS-DA thereby overcomes the problem of multicollinearity and simultaneously tries to maximize group differences, which was the most effective strategy in our simulations. Nevertheless, differences between methods only became apparent when we simulated huge treatment effects (Cohen's d equal to 2.0). However, in our practical example of combining ordination techniques with statistical testing methods to investigate treatment effects, PLS-DA managed to identify all variables with simulated treatment effects as important for the observed multivariate pattern. Follow-up statistical tests did not capture all differences successfully. PLS-DA might therefore be a useful strategy to preselect important endpoints for targeted statistical testing with the goal of reducing the overall number of tests.

## Conclusion

Assessing therapeutic success in preclinical neurotrauma studies remains challenging when small samples are combined with small effect sizes. Our simulation study demonstrated that linear mixed effects models offer a slightly increased power in case of equal variance whereas Welch's ANOVA should be used when homogeneity of variance is not present. Additionally, PLS-DA offers a higher sensitivity to detect treatment effect patterns than PCA and RDA, whereas classical LDA leads to overfitting and false inferences in multivariate data sets with few measurements per group. Although we based our simulation on a real neurotrauma preclinical study, our findings might be more generally applicable to multivariate data sets with a similar correlation structure as we applied standardized measures of effect sizes which are not restricted to a specific endpoint or type of study.

Ultimately, translational success of animal trials in neurotrauma would greatly benefit from appropriate sample size calculation prior to conduct of the study. When this is not feasible, it is advantageous to re-evaluate estimates of treatment effect with combined evidence from existing studies (if available) by performing systematic reviews and meta-analyses.

## Supporting information

**S1 Appendix. The file contains the mean and variance vector of the simulated control group and the correlation matrix used to sample data from multivariate distributions under different simulation scenarios.** Figs 1–4 show comparisons of type I error rate and empirical power of the four different multivariate statistics used to evaluate the significance of MANOVA tests.
(PDF)

**S2 Appendix. Comparison of ordination techniques to detect treatment effect patterns when no treatment effects were simulated.** The file contains the results from 1000 simulated data sets without treatment effects, 5 measurements per group with data obtained from a multivariate normal distribution with equal variance in all groups. LDA, PCA, RDA, or PLS-DA were considered to falsely capture a non-existing treatment effect pattern if the 95% confidence ellipse around the control group did not overlap with the confidence ellipses of treatment groups (dose1 to dose3).
(PDF)

**S3 Appendix. Comparison of ordination techniques to detect treatment effect patterns with huge simulated treatment effects (Cohen's d equal to 2.0).** The file contains results from 1000 simulated data sets with 5 measurements per group and data obtained from a

multivariate normal distribution with equal variance in all groups. PCA, RDA, or PLS-DA were considered to correctly capture a treatment effect pattern if the 95% confidence ellipse around the control group did not overlap with the confidence ellipses of the treatment groups (dose 1 to dose3).
(PDF)

## Author Contributions

**Conceptualization:** Susanne Gerber.

**Data curation:** Hristo Todorov.

**Formal analysis:** Hristo Todorov.

**Investigation:** Hristo Todorov.

**Methodology:** Hristo Todorov.

**Supervision:** Susanne Gerber.

**Validation:** Emily Searle-White.

**Visualization:** Hristo Todorov.

**Writing – original draft:** Hristo Todorov.

**Writing – review & editing:** Emily Searle-White, Susanne Gerber.

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
