## [Decision Letter · Decision Letter 0]

11 Feb 2020

PONE-D-19-16128

Applying univariate vs. multivariate statistics to investigate therapeutic efficacy in controlled preclinical neurotrauma trials: A Monte Carlo simulation study

PLOS ONE

Dear Dr Gerber,

Thank you for submitting your manuscript to PLOS ONE. After careful consideration, we feel that it has merit but does not fully meet PLOS ONE’s publication criteria as it currently stands. Therefore, we invite you to submit a revised version of the manuscript that addresses the points raised during the review process.

In particular, one of the reviewers brought up substantive objections to the approach outlined in your work; it would be particularly helpful for you to address those concerns directly. Furthermore, other reviewers asked to clarify point of methodology in the abstract and main text of the manuscript. Please be sure to address those as well. 

We would appreciate receiving your revised manuscript by Mar 27 2020 11:59PM. To enhance the reproducibility of your results, we recommend that if applicable you deposit your laboratory protocols in protocols.io, where a protocol can be assigned its own identifier (DOI) such that it can be cited independently in the future. For instructions see: http://journals.plos.org/plosone/s/submission-guidelines#loc-laboratory-protocols

A rebuttal letter that responds to each point raised by the reviewer(s). This letter should be uploaded as separate file and labeled 'Response to Reviewers'.A marked-up copy of your manuscript that highlights changes made to the original version. This file should be uploaded as separate file and labeled 'Revised Manuscript with Track Changes'.An unmarked version of your revised paper without tracked changes. This file should be uploaded as separate file and labeled 'Manuscript'.

Please note while forming your response that, if your article is accepted, you may have the opportunity to make the peer review history publicly available. The record will include editor decision letters (with reviews) and your responses to reviewer comments. If eligible, we will contact you to opt in or out.

We look forward to receiving your revised manuscript.

Kind regards,

Marco Bonizzoni, Ph.D.

Academic Editor

PLOS ONE

'The work of HT was funded by Fresenius Kabi Deutschland GmbH. The work of ESW was funded by the Center for Computational Sciences in Mainz (CSM). The work of SG was partly supported by the CRC 1193.'

Additionally, because some of your funding information pertains to commercial funding, we ask you to provide an updated Competing Interests statement, declaring all sources of commercial funding.

In your Competing Interests statement, please confirm that your commercial funding does not alter your adherence to PLOS ONE Editorial policies and criteria by including the following statement: "This does not alter our adherence to PLOS ONE policies on sharing data and materials.” as detailed online in our guide for authors  http://journals.plos.org/plosone/s/competing-interests.  If this statement is not true and your adherence to PLOS policies on sharing data and materials is altered, please explain how.

Please include the updated Competing Interests Statement and Funding Statement in your cover letter. We will change the online submission form on your behalf.

3. Thank you for providing the following Funding Statement: 

We note that one or more of the authors is affiliated with the funding organization, indicating the funder may have had some role in the design, data collection, analysis or preparation of your manuscript for publication; in other words, the funder played an indirect role through the participation of the co-authors.

If the funding organization did not play a role in the study design, data collection and analysis, decision to publish, or preparation of the manuscript and only provided financial support in the form of authors' salaries and/or research materials, please review your statements relating to the author contributions, and ensure you have specifically and accurately indicated the role(s) that these authors had in your study in the Author Contributions section of the online submission form. Please make any necessary amendments directly within this section of the online submission form.  Please also update your Funding Statement to include the following statement: “The funder provided support in the form of salaries for authors [insert relevant initials], but did not have any additional role in the study design, data collection and analysis, decision to publish, or preparation of the manuscript. The specific roles of these authors are articulated in the ‘author contributions’ section.”

If the funding organization did have an additional role, please state and explain that role within your Funding Statement.

Please also provide an updated Competing Interests Statement declaring this commercial affiliation along with any other relevant declarations relating to employment, consultancy, patents, products in development, or marketed products, etc.  

Reviewers' comments:

Reviewer's Responses to Questions

**Comments to the Author**

1. Is the manuscript technically sound, and do the data support the conclusions?

Reviewer #1: Yes

Reviewer #2: No

Reviewer #3: Yes

2. Has the statistical analysis been performed appropriately and rigorously? 

Reviewer #1: Yes

Reviewer #2: No

Reviewer #3: Yes

5. Review Comments to the Author

Reviewer #1: Thank you for inviting me to review the manuscript PONE-D-19-1628 entitled “Applying univariate vs. multivariate statistics to investigate the therapeutic efficacy in controlled preclinical neurotrauma trials: A Monte Carlo simulation study”. Motivated by a clinical trial, the manuscript studied the empirical power of different statistics analyzing the trial data with multiple correlated endpoints with repeated measures. In general ,the manuscript was well written and has many merits: addressing a real clinical issue, conducting a simulation study, providing good ground for manipulated factors, visual examination of treatment effect, clear description of software used providing reproducibility options, good definition of the evaluating criteria of assessing the models and statistics. There are some improvements needed before it’s ready for publication.

Abstract

It is not clear what “acceptable level” of power means.

What does “20 measurements per group” mean? Does it mean 20 subjects per group?

Methods

In general the simulation scenario needs to be clarified: did the author generated 4 groups (1 control, 3 treatments of different doses) and 7 endpoints each with 3 times of measurements (page 5)? Or did the authors generated 2 groups of 1 control and 1 treatment as Hotelling’s T square was used in the MANOVA analysis (page 10). If the simulation is in 4 groups, how is effect size Cohen’s d defined? Is it defined as difference between each two groups?

In general, how the MANOVA multivariate responses are defined is not clear: is it the different time points of a specific endpoints that are treated are multivariate (page 7)? Or is it the 7 different endpoints that are treated as multivariate for a specific time? From Appendix 1, it looks like both the endpoints and time points are used as different response, altogether 19 columns. Please clarify in the method section.

Simulation procedure: the way simulation was done is a good representation of the clinical trial by bootstrapping method, yet it can also make the results of the study biased to a specific trial and limit its generalizability.

Simulation factors: How was the simulation scenarios of 24 defined? There were 4 levels in n, 4 levels in ES (0, .2, .5, .8), 3 levels in variance. If effect size = 0 is not treated as a simulation condition, details need to be elaborated. What about the distribution of dependent variables that also include log transformation?

Multivariate dimensionality reduction techniques for pattern analysis: please explain why is huge treatment effects (Cohen’s d=2.0) chosen as an example.

Reviewer #2: This paper seeks to encourage potentially more appropriate analysis of data from preclinical experiments involving multiple outcomes and multiple experimental groups. The main hypothesis is that a multivariate consideration of the outcomes rather than multiple univariate tests may be more powerful and the goal is to identify which multivariate method might be most useful via a simulation study.

The goal of the paper is laudable, but there are several issues with the approach and holes that limit the validity of the conclusions.

The first set of issues is related to the comparison of univariate to multivariate tests with respect to type I error. The way empirical type I error is calculated is ill-conceived. As the authors state, the univariate tests will maintain close to nominal levels of type I error on a test-by-test basis. In practice, the concern would be for the case when there really is no effect, but a few outcomes have (unadjusted) p<0.05. That should be the comparison. How often does a set of univariate tests give a "wrong result" in terms of concluding the treatment has an effect based on one or more significant p-values (if that is the rule for finding a significant difference). But if one were to accept 1 or more significant p-values among any of the multiple tests as indicating difference, standard practice would be to adjust the p-values using a Bonferroni adjustment or some other method of controlling the type I error rate. There is a similar issue with how power is calculated.

I am also confused by the combination of repeated measurement of multiple outcomes into a single vector without taking any of that information into account in most of the multivariate analyses. In this setting, it would be (somewhat) uncommon to do univariate on all items separately. Mixed effects regression or repeated measures ANOVA would be the choice to make, and mixed effects models for multivariate repeated measures do exist. It would have been helpful to consider these alternatives. It would seem that ignoring knowledge about the data structure that comes from the experimental design might also severely hamper the performance of the PCA ANOVA and dimensionality reduction techniques.

Another issue with multivariate methods that is not addressed is missing data. MANOVA cannot handle data that are missing at random while mixed effects models can. Although not a focus of this analysis, the limitations of MANOVA in this regard would suggest opting for a more flexible method like mixed effects models for comparative purposes.

Perhaps the most important question is whether multivariate techniques, even if they improve power in some modest way, are of value in the preclinical setting since, as the authors state in the introduction, they have "increased complexity of analysis and interpretation of results." If separate modeling of each outcome provides an accurate representation of the effect of the treatment on that outcome, does a multivariate p-value or a data reduction technique help if we can't easily interpret the effect? Perhaps coupling a data analysis example to the simulations where all of the methods were applied to a real data set would help to clarify the analytic methods that were actually applied and the issues in interpretation that come with the methods.

Reviewer #3: The paper evaluates the performance of univariate ANOVA and Welch’s ANOVA tests versus multivariate techniques based on the simulation study, taking into account sample size/effect size, normality and homogeneity of variance. The idea makes sense intuitively (according to the statistical textbook/theory) and the result may be helpful for some researchers in application. However, the methodology is not novel and the broadness of the application may be not enough. It may be helpful to medical researchers. I have some concerns and comments as follows.

(1) I assume that this is more like a statistical research paper, not medical research paper

(2) It is not clear why does the title of this paper include “in controlled preclinical neurotrauma trials”? It seems that the result of the simulation study can be applied to different trials (or clinical trials), not just neurotrauma trials only.

(3) The results from the simulation study show that Welch’s ANOVA is as powerful as classical ANOVA tests with variance homogeneity and outperformed the remaining methods when this assumption was violated. However, most animal data are much more homogeneous (less variation), in comparison with the clinical data (human being). That is to say, the result from the simulation study may be helpful to a small clinical study (bigger variation), not just preclinical trials (smaller variation).

(4) In simulation factors section, a sample size of 5 – 20 may be too small. It will be interesting to see more scenarios with a ranged from 5 – 50 (say) to benefit more people (similar to my comment (2))

(5) In simulation factors section, the correlation in multivariate normal distribution of dependent variables is missing (which is important)

(6) In the simulation study, you may generate the data from other distributions (not normal/ log-normal distribution. This can be another factor in the simulation study. For example, a gamma distribution or Weibull distribution.

6. PLOS authors have the option to publish the peer review history of their article (what does this mean?). If published, this will include your full peer review and any attached files.

Reviewer #1: No

Reviewer #2: No

Reviewer #3: No

---

## [Author Response · Author response to Decision Letter 0]

6 Mar 2020

Dear Editor, Dear Reviewers,

we would like to take the opportunity to thank you for giving us the chance to revise our Manuscript and for your very valuable comments giving us the chance to improve our work.

We have addressed all the reviewers’ comments. A point-by-point response to the individual reviewers’ questions is provided below.

Please find the comments also in our "Letter of response" in a clearer format and with highlighted paragraphs.

Sincerely,

Susanne Gerber,

on behalf of the authors.

Reviewer #1

Abstract

It is not clear what “acceptable level” of power means.

Usually, the acceptable level of power (also the value used for sample size estimations) is 80%. We have adapted the text to make this understandable.

What does “20 measurements per group” mean? Does it mean 20 subjects per group?

In the course of the manuscript, we have used the terms “measurements per group”, “subjects per group” and “replicates per group” interchangeably. We have included a statement in the methods section to make this more clear.

Methods

In general the simulation scenario needs to be clarified: did the author generated 4 groups (1 control, 3 treatments of different doses) and 7 endpoints each with 3 times of measurements (page 5)? Or did the authors generated 2 groups of 1 control and 1 treatment as Hotelling’s T square was used in the MANOVA analysis (page 10). If the simulation is in 4 groups, how is effect size Cohen’s d defined? Is it defined as difference between each two groups?

We apologize for these ambiguities. We simulated 4 groups in each case, one of these groups was considered to be a control group and the remaining 3 groups were considered to be treatment groups. The effect size was always considered as the difference between the control group and each of the treatment groups. 

In the statistical analysis, all 4 groups were considered simultaneously. In the MANOVA analysis, we actually used the Lawley-Hotelling-statistic which is a generalization of the Hotelling’s T square statistic and is calculated as trace(E-1H) where E denotes the error matrix and H denotes the hypothesis matrix. Alternatively, the Lawley-Hotelling statistic is also equal to the sum of the eigenvalues of the (E-1H)-matrix. We have updated the name of this test statistic to Lawley-Hotelling trace in the revised version of the manuscript to avoid a possible confusion.

In general, how the MANOVA multivariate responses are defined is not clear: is it the different time points of a specific endpoints that are treated are multivariate (page 7)? Or is it the 7 different endpoints that are treated as multivariate for a specific time? From Appendix 1, it looks like both the endpoints and time points are used as different response, altogether 19 columns. Please clarify in the method section.

Our data set included 6 variables, each of these 6 variables was measured at three different time points. A seventh variable was measured only once, but we have excluded it from the revised manuscript as we investigated additional methods for repeated measures. A separate MANOVA test was performed for each endpoint. The repeated measures at the three time points served as dependent variables for each MANOVA test. Thus, we performed 6 MANOVA tests with three dependent variables each. In the univariate ANOVA tests, each repeated measure was considered as a separate variable.We have tried to make this more clear in the revised version of the manuscript.

Simulation procedure: the way simulation was done is a good representation of the clinical trial by bootstrapping method, yet it can also make the results of the study biased to a specific trial and limit its generalizability.

We thank the reviewer for this important comment. We agree that our approach might limit the generalizability of the results. Since there are countless options for a number of variables, mean vector and correlation structure for a multivariate data set, we decided to base the simulation procedure on a real study, in order to have simulated data as realistic as possible. In order to increase the generalizability, we did not directly use the mean vector and correlation matrix of the original data but used the bootstrap procedure which should at least give us more population specific and not just sample specific estimates of the parameters used for the simulations.

Simulation factors: How was the simulation scenarios of 24 defined? There were 4 levels in n, 4 levels in ES (0, .2, .5, .8), 3 levels in variance. If effect size = 0 is not treated as a simulation condition, details need to be elaborated. What about the distribution of dependent variables that also include log transformation?

We thank the reviewer for pointing out this inconsistency. When calculating the number of simulation scenarios we mistakenly overlooked the factor of sample size which resulted in this incorrect number. In the revised version of the manuscript, we have performed additional simulations with data following a multivariate gamma distribution and sample sizes of 30, 40 and 50 subjects per group. Therefore, we now actually have 252 scenarios (with 3 different distributions, 4 different effect sizes, 3 different variance ratios and 7 different sample sizes per group).

Multivariate dimensionality reduction techniques for pattern analysis: please explain why is huge treatment effects (Cohen’s d=2.0) chosen as an example.

In this example, we simulated 5 subjects per group and Cohen’s d = 2.0 was the lowest value for the effect size for which we observed a difference between the ordination methods, indicating that smaller effect sizes cannot be detected given a sample size of 5 subjects per group. We have included this in the revised manuscript.

Reviewer #2

The first set of issues is related to the comparison of univariate to multivariate tests with respect to type I error. The way empirical type I error is calculated is ill-conceived. As the authors state, the univariate tests will maintain close to nominal levels of type I error on a test-by-test basis. In practice, the concern would be for the case when there really is no effect, but a few outcomes have (unadjusted) p<0.05. That should be the comparison. How often does a set of univariate tests give a "wrong result" in terms of concluding the treatment has an effect based on one or more significant p-values (if that is the rule for finding a significant difference). But if one were to accept 1 or more significant p-values among any of the multiple tests as indicating difference, standard practice would be to adjust the p-values using a Bonferroni adjustment or some other method of controlling the type I error rate. There is a similar issue with how power is calculated.

We believe we used the standard approach to calculate the empirical type I error rate and power of the different tests. Furthermore, we believe our strategy complies with what the reviewer described as to how type I error rate and power should be estimated. We determined type I error rate when no treatment effects were simulated as in this case no significant differences should be detected. For the univariate tests, for example, we performed 18 ANOVA tests in each simulation round (repeated 1000 times under each simulated scenario). Each time, we determined the fraction of these 18 tests that were significant and finally reported the average fraction over the 1000 simulations as the final estimate of type I error rate. This is mathematically equivalent to counting the number of significant tests and dividing that by the number of total tests performed. For MANOVA tests, for example, we performed 6 MANOVAs each time and determined the average fraction of these MANOVA tests that were significant over the 1000 simulations. We believe that normalizing the type I error rate and power to the number of tests performed is the only objective way to compare univariate and multivariate methods. Otherwise, the difference in type I error rate or power would be attributable to the difference in the number of tests performed.

In practice, a p-value adjustment method would indeed be applied to control the family wise error rate or the false discovery rate. In our simulations, we did not apply a p-value adjustment, because we knew the ground truth. Therefore, each time a p-value was below 0.05 we knew if this was correct depending on the treatment effect we simulated. We hope our strategy is now more understandable and acceptable to the reviewer. 

I am also confused by the combination of repeated measurement of multiple outcomes into a single vector without taking any of that information into account in most of the multivariate analyses. In this setting, it would be (somewhat) uncommon to do univariate on all items separately. Mixed effects regression or repeated measures ANOVA would be the choice to make, and mixed effects models for multivariate repeated measures do exist. It would have been helpful to consider these alternatives. It would seem that ignoring knowledge about the data structure that comes from the experimental design might also severely hamper the performance of the PCA ANOVA and dimensionality reduction techniques.

We thank the reviewer for this important comment. We believe that it is not that uncommon in the field of animal studies to perform separate ANOVA tests for each time point for repeated measures variables. Furthermore, while the endpoints in the original study we based our simulations on, were measured repeatedly, in the context of our simulations this more generally translates to endpoints which follow a certain correlation structure. Thus, the endpoints which were originally measured repeatedly correspond to variables which are more strongly correlated among each other than with other variables. For this reason, we believe it is a legitimate approach to analyze them using separate ANOVA tests or MANOVA tests in our simulations.

Nevertheless, in practice, is does seem imprudent to ignore the repeated measure nature of data. Therefore, we have also included linear mixed effects analysis of repeated measures in the revised version of the manuscript. Since the original data included one endpoint measured only once, we have excluded it from the updated analysis.

Regarding the dimensionality reduction techniques, we believe that their performance is not severely limited by repeated measures data. The assumption is that repeated measures are correlated with each other and since the dimensionality reduction techniques are calculated based on correlation matrices, this information is implicitly taken into account. For instance, variables with repeated measures often load on the same component in the ordination procedure.

Another issue with multivariate methods that is not addressed is missing data. MANOVA cannot handle data that are missing at random while mixed effects models can. Although not a focus of this analysis, the limitations of MANOVA in this regard would suggest opting for a more flexible method like mixed effects models for comparative purposes.

We agree with the reviewer that missing data might pose a significant limitation for multivariate techniques. If the level of missingness is not great (e.g. less than 10%) and the missing at random condition is satisfied, then these missing values might be imputed and a multivariate technique still applied. If not, a more flexible method such as linear mixed effects models would be the natural choice. While a systematic investigation of different degrees of missingness and imputation techniques is beyond the scope of our current study, we have included these considerations in the discussion of the revised manuscript.

Perhaps the most important question is whether multivariate techniques, even if they improve power in some modest way, are of value in the preclinical setting since, as the authors state in the introduction, they have "increased complexity of analysis and interpretation of results." If separate modeling of each outcome provides an accurate representation of the effect of the treatment on that outcome, does a multivariate p-value or a data reduction technique help if we can't easily interpret the effect? Perhaps coupling a data analysis example to the simulations where all of the methods were applied to a real data set would help to clarify the analytic methods that were actually applied and the issues in interpretation that come with the methods.

We agree with the reviewer that our results indicate that multivariate techniques such as MANOVA do not offer a practical benefit in preclinical studies for formal statistical testing as our results indicated that the gain in power compared to ANOVA tests was not nearly sufficient enough to justify the complexity of interpretation. Furthermore, in the updated simulations, linear mixed effects models outperformed the remaining methods with repeated measures data. We have updated the discussion and conclusion to reflect these findings. 

However, we believe that dimensionality reduction techniques are useful beyond formal hypothesis testing for data exploration purposes. We have tried to demonstrate this by including a practical data analysis example with one simulated data set in the revised version of the manuscript (Fig. 7 and table 1). When we simulated treatment effects on only half of the variables, PLS-DA captured the multivariate pattern in the data and managed to identify the variables with simulated treatment effects as important for group separation in reduced multivariate space. We hope this applied example demonstrates the usefulness of these methods for data exploratory purposes whereas we suggest that linear mixed effects models or ANOVA tests are more appropriate for formal hypothesis testing.

Reviewer #3

(1) I assume that this is more like a statistical research paper, not medical research paper

Our goal in the current study was to systematically compare the performance of a number of univariate and multivariate techniques by manipulating factors justified by the statistical assumptions of the different tests. As such, our investigations are more statistical in nature. While the practical implications of our results could hopefully be useful to the medical research community when confronted with similar data, we believe that applying especially the multivariate techniques requires a solid statistical background which pure experimentalists might lack. Thus, our results would hopefully be helpful to biostatisticians and data analysts who also have a good understanding of the theoretical background of the statistical methods.

(2) It is not clear why does the title of this paper include “in controlled preclinical neurotrauma trials”? It seems that the result of the simulation study can be applied to different trials (or clinical trials), not just neurotrauma trials only.

We thank the reviewer for acknowledging the potential of our study to have a broader application than simply preclinical neurotrauma studies. We chose to be more cautious and limit ourselves to this type of studies as we based the simulations on data from a real neurotrauma study. However, since we used standardized measures for treatment effects, our findings could theoretically have broader applications. We have discussed this in the revised version of the manuscript and also slightly changed the title.

(3) The results from the simulation study show that Welch’s ANOVA is as powerful as classical ANOVA tests with variance homogeneity and outperformed the remaining methods when this assumption was violated. However, most animal data are much more homogeneous (less variation), in comparison with the clinical data (human being). That is to say, the result from the simulation study may be helpful to a small clinical study (bigger variation), not just preclinical trials (smaller variation).

We agree with the reviewer that our study results might be useful to a small clinical trial where data are usually more heterogenous than animal data. Furthermore, we have implemented the reviewer’s suggestion to simulate bigger group sizes (e.g. 30, 40 and 50 replicates per group) and we have also included simulations with data coming from a gamma distribution thereby hopefully increasing the impact of our findings. 

(4) In simulation factors section, a sample size of 5 – 20 may be too small. It will be interesting to see more scenarios with a ranged from 5 – 50 (say) to benefit more people (similar to my comment (2))

As the reviewer helpfully suggested, we have extended the simulation procedure in the revised manuscript to also include simulations with 30, 40 and 50 subjects per group in order to increase the impact of our research and hopefully provide beneficial results to a broader circle of researchers. 

(5) In simulation factors section, the correlation in multivariate normal distribution of dependent variables is missing (which is important)

We thank the reviewer for this observation. We have included a reference to where the correlation matrix of the dependent variables can be found in the simulation factors section (S1 Appendix Table 2).

(6) In the simulation study, you may generate the data from other distributions (not normal/ log-normal distribution. This can be another factor in the simulation study. For example, a gamma distribution or Weibull distribution.

According to this helpful suggestion, we have also included simulations with data coming from a multivariate gamma distribution in the revised version of the manuscript.

---

## [Editor Report · Decision Letter 1]

10 Mar 2020

Applying univariate vs. multivariate statistics to investigate therapeutic efficacy in (pre)clinical trials: A Monte Carlo simulation study on the example of a controlled preclinical neurotrauma trial.

PONE-D-19-16128R1

Dear Dr. Gerber,

After evaluation of the revised version of your manuscript you recently submitted, we are pleased to inform you that your manuscript has been judged scientifically suitable for publication and will be formally accepted for publication once it complies with all outstanding technical requirements.

With kind regards,

Marco Bonizzoni, Ph.D.

Academic Editor

PLOS ONE

---

## [Editor Report · Acceptance letter]

12 Mar 2020

PONE-D-19-16128R1 

Applying univariate vs. multivariate statistics to investigate therapeutic efficacy in (pre)clinical trials: A Monte Carlo simulation study on the example of a controlled preclinical neurotrauma trial. 

Dear Dr. Gerber:

I am pleased to inform you that your manuscript has been deemed suitable for publication in PLOS ONE. Congratulations! Your manuscript is now with our production department. 

With kind regards,

on behalf of

Dr. Marco Bonizzoni 

Academic Editor

PLOS ONE